# Fat Infiltration of Multifidus Muscle Is Correlated with Neck Disability in Patients with Non-Specific Chronic Neck Pain

**DOI:** 10.3390/jcm11195522

**Published:** 2022-09-21

**Authors:** Francis Grondin, Sébastien Freppel, Gwendolen Jull, Thomas Gérard, Teddy Caderby, Nicolas Peyrot

**Affiliations:** 1Laboratory IRISSE, EA4075, Faculty of Human and Environment Sciences, University of La Réunion, 97430 Le Tampon, France; 2Neurosurgey Department, University Hospital of La Réunion, 97410 Saint-Pierre, France; 3Physiotherapy, School of Health and Rehabilitation Sciences, The University of Queensland, Brisbane 4072, Australia; 4Institute of Health Engineering, University of Picardie Jules Verne, 80000 Amiens, France; 5Laboratory Movement Interactions Performance, MIP UR4334, Le Mans University, 72000 Le Mans, France

**Keywords:** chronic neck pain, fat infiltration, neck disability, multifidus, correlation

## Abstract

**Background:** Chronic non-specific neck pain (CINP) is common, but the etiology remains unclear. This study aimed to examine the relationship between cervical muscle composition (cervical multifidus and longus capitis/longus colli), morphometry, range of movement, muscle function, and disability severity (Neck Disability Index) in patients with CINP. **Methods:** From September 2020 to July 2021, subjects underwent cervical MRI and clinical tests (cervical range of motion, cranio-cervical flexion test, neck flexor, and extensor muscle endurance). MRI analysis comprised muscle cross-sectional area, volume, and fat infiltration of multifidus and longus colli between C4 and C7 levels. **Results:** Twenty-five participants were included. Multiple linear regression analysis indicated that NDI was positively correlated with the volume percentage of fat infiltration of the multifidus (B = 0.496), negatively correlated with fat-free muscle volume of the multifidus normalized by subject height (B = −0.230), and accounted for 32% of the variance. There was no relationship between neck disability and longus capitis/longus colli morphology. We also found no relationship between neck disability scores, neck flexor or extensor muscle endurance, or the outcome motor control test of craniocervical flexion (*p* > 0.05). **Conclusions:** Neck disability was moderately correlated with the percentage of fat volume in the multifidus muscle and fat-free volume of the multifidus. There was no relationship between NDI scores and muscle function test outcomes or any fat or volume measures pertaining to the longus colli muscle.

## 1. Introduction

Neck pain is common in the general population [1] and, as reported in the Global Burden of Disease Study, is a leading cause of disability worldwide [2], requiring special attention from healthcare providers and researchers [3]. Non-specific or idiopathic neck pain is the most common disorder [4]. Although the etiology of chronic non-specific neck pain (CINP) remains unclear, recent systematic reviews recommend conservative treatment. Specific exercises for the neck and axioscapular muscles are beneficial for these patients [5,6]. Hence, several studies have investigated neck muscle impairments in patients with CINP to guide specific and effective exercises.

Several impairments have been identified in neck flexor motor control. In the motor control task of craniocervical flexion, activity levels of the deep neck flexors (longus capitus/longis colli) were shown to be reduced in patients with CINP, while activity in the superficial flexors was increased compared to controls [7,8]. Delayed onsets of the cervical flexor and extensor muscles have also been recorded during arm movements and body perturbation tasks in these patients [7,9]. Reductions in strength production and endurance occur in both the craniocervical and cervical flexors and cervical extensors in people with CINP [10,11,12,13]. CINP-related alterations in cervical muscle morphometry and composition have also been revealed. Specifically, a reduction in size of the multifidus, semispinalis cervicis [14,15], and the deep longus colli muscles [15,16] have been identified using ultrasound imaging. Reduced cross-sectional area (CSA) [17,18] and volume [19] of multifidus and semispinalis cervicis have been measured with magnetic resonance imaging (MRI). In contrast, no changes in size were found in the superficial neck muscles (semispinalis capitis, splenius capitis) [14,20] or sternocleidomastoid [20] with ultrasound imaging (US).

Despite evidence of muscle function alterations, morphometry, and composition in patients with CINP, there is little information about the relationship between impairments of these muscles and patient-reported neck disability or pain. Understanding the relationship between precise neck muscle impairments and disability might further help target specific exercises for appropriate management of CINP. Previous studies have produced mixed results regarding associations between pain or disability and deep neck flexor, neck flexor, and extensor muscle endurance [11,20,21,22]. From a morphological perspective, weak negative correlations have been demonstrated between pain and US measures of multifidus thickness [14,23], as well as disability with semispinalis cervicis muscle thickness [14].

MRI presents the best and most precise non-invasive examination of muscle morphometry and composition [24]. Of the three studies that have assessed neck muscle morphometry in patients with CINP, two assessed neck muscle CSA [17,18] and one measured neck muscle volume [19]. To date, no study has assessed the volume of fat infiltration in neck muscles and its association with disability severity in this patient group. Recently, MRI-based studies have quantified the pathobiology of changes in muscle morphometry (atrophy) and composition (fatty infiltration) in patients with other neck disorders [25,26,27]. Significantly higher fat infiltration has been demonstrated in the multifidus of patients with degenerative cervical myelopathy [25] and with recalcitrant whiplash-associated disorders [26]. Fatty infiltration has been associated with poor outcomes (on the basis of the Neck Disability Index) for patients with whiplash-associated disorders [28,29], as well as with poor functional recovery after surgery among people with cervical degenerative myelopathy [30]. Patients with whiplash-associated disorders with severe disability had 45% greater fat infiltration compared to those with mild to moderate disability [31]. Furthermore, those who recovered had significantly less neck muscle fat infiltration in the multifidus muscle [26]. There is little knowledge, however, of any relationship between MRI measures of neck muscle morphology and composition and disability in patients with CINP that could improve our understanding of this burdensome disorder for diagnosis and management.

This study aimed to examine the relationship between disability severity evaluated by the Neck Disability Index (NDI) and cervical muscle composition, morphometry, and function (atrophy, fat infiltration, motor control, and muscle endurance) in patients with CINP. The deep cervical flexor (longus colli) and extensor (multifidus) muscles were targeted as composition and morphometry changes have been found in these muscles in other neck disorders. We hypothesized that the significant part of the NDI score variance would be explained by muscle composition and morphometry (volume, fat infiltration) and muscle function (motor control and muscle endurance).

## 2. Materials and Methods

### 2.1. Participants

Participants were selected from consecutive patients referred to the Neurosurgery Department, Centre Hospitalier Universitaire Réunion, France, from September 2020 to July 2021 (La Reunion, France). They were included if aged between 18 and 65 years and had non-specific neck pain for at least 3 months (chronic) with symptoms provoked by neck movement. Patients were excluded if they presented a history of head or neck surgery, whiplash, or neurologic features; neck or thoracic fracture; cancer; rheumatoid arthritis; drug or alcohol abuse; or recent neck, shoulder, or thoracic rehabilitation.

All subjects gave their informed consent for inclusion before they participated in the study. This observational cross-sectional study was approved by the local ethics committee (Comité de Protection des Personnes Sud Est V-No. 20.04.09.51157). 

### 2.2. Experimental Procedure

Participants provided demographic information and completed the French version of the Neck Disability Index (NDI) [32]. The NDI is the most widely used assessment tool measuring disability in patients with acute and chronic neck pain or neck injury. It has been shown as an efficient and trustworthy tool to measure and monitor neck-related disability [33]. The NDI contains 10 items: pain, personal care, lifting, reading, headaches, concentration, work, driving, sleeping, and recreation. Each item is scored on a 0 to 5 rating scale. The NDI is scored with points summed/50 and expressed as a percent: 0–4 points (0–8%) no disability, 5–14 points (10–28%) mild disability, 15–24 points (30–48%) moderate disability, 25–34 points (50–64%) severe disability, and 35–50 points (70–100%) complete disability.

Participants also completed the 12-Item Short Form Health Survey (SF-12) [34], the Pain Catastrophizing Scale, and the Numeric Pain Rating Scale [35]. One physiotherapist who was blind to the initial assessment and imagery of each participant (NDI, pain, MRI) performed all clinical tests, which included the craniocervical flexion test (CCFT), neck flexor and neck extensor muscle endurance tests, and measures of cervical range of motion.

### 2.3. Cranio-Cervical Flexion Test

CCFT evaluates deep neck flexor (longus colli, longus capitis) activation and low-load endurance capacity. The test was performed in supine position, as previously described by Jull et al. [8]. Patients were asked to attempt five progressives inner range contractions guided by feedback from a pressure sensor placed behind their neck and pre-inflated to a baseline of 20 mmHg (Chattanooga Stabilizer Group Inc., Hixson, TN, USA) (Figure 1). Participants were asked to nod the chin (as if saying “yes”) to reach each 2 mmHg incremental target from 22 mmHg to 30 mmHg, and to hold the position for 10 s. Craniocervical flexion for each increment was repeated 3 times. The test was stopped if the patient was unable to hold the position, if they performed a retraction action, or if holding the position was jerky. The pressure level at which the participant could control the contraction for three repetitions was documented for analysis. CCFT has excellent intrarater reliability with an intra correlation class coefficient (ICC) of 0.98 and interval of confidence of 95% (95%IC) ranged between 0.95 and 0.90 [36], with good validity [37,38], with standard error of measurement less than 1.7 mmHg [38].

### 2.4. Neck Flexor and Extensor Muscle Endurance

Muscle endurance tests were performed as previously described for the neck flexor [39] and extensor muscles [40]. For the neck flexor test, participants were supine. They performed maximal upper cervical flexion by tucking the chin to the throat and then lifting the head up 2 cm. The examiner placed their hand between the bed and back of the head. Participants had to hold the position for as long as possible. The test ended if the upper cervical flexion position was lost (the chin lifted), the participant’s head touched the clinician’s hand for more than 1 s [41], or the participant was unable to hold the head position. This test has excellent reliability with an ICC of 0.93 (95%IC: 0.86–0.97) and a standard error of measurement (SEM) of 6.4 seconds (s) [39].

For the neck extensor endurance test, participants were positioned in prone, lying with their head extended off the bed [42]. Arms were alongside their body, and the trunk was stabilized with a belt positioned over the scapular region. A 2 kg weight was suspended from the head. The instruction was to tuck the chin and maintain the head position for as long as possible. The test stopped when the weight touched the ground for more than 2 s [42]. This test has moderate intra-examiner reliability with ICC of 0.88 (95%IC: 0.75–0.95) [39].

### 2.5. Cervical Range of Motion

Cervical range of motion in flexion/extension, lateral flexion, and rotation were measured using an iPhone 7 (Apple Inc., Cupertino, CA, USA) fixed on the patient’s head. The procedure was undertaken as previously described by Guidetti et al. [43]. This method has excellent reliability (ICC’s range from 0.998 to 0.999 (95%CI: 0.996–0.999)) [43].

### 2.6. Muscle Cross-Sectional Area, Volume, and Fat Infiltration

MRI acquisition of muscle morphological measures was obtained using a conventional spin-echo pulse sequence, time repetition of 1250 ms, time echo of 123 ms, resolution 256 × 236, and field of view (FOV) 100%, with a time acquisition of 2 min and 3 s (Simens, Erlangen, Germany). Measures of multifidus and longus colli CSAs were taken from a T2-weighted axial MR image using Osirix (Version 1.43, National Institutes of Health, Bethesda, MD, USA) (Figure 2). Two examiners who were blind to the conditions of the participants and other clinical parameters measured the cross-sectional areas of the longus colli and multifidus muscles. The agreement of the two examiners was considered for analysis. Measurements were obtained bilaterally at the mid-disc level of C3/4, C4/5, C5/6, and C6/7. This method of CSA measurement has excellent intra-rater reliability with ICC of 0.96 (95%CI: 0.91–0.98) for novices and 0.99 (95%CI: 0.98–0.99) for experts [24,44].

Fat infiltration was measured by semi-automatic selection of a threshold signal within the total muscle CSA, which included only pixels corresponding to non-fat signals. Because of the heterogeneity of the fat signal intensity between participants, threshold was determined by minimal pixel intensity of posterior subcutaneous fat measured at C4/5, C5/6, C5/6, and C6/7 levels (Figure 2). This thresholding technique has been previously described [24]. Measurements of interest included fat-free muscle CSA of longus colli (rCSA-Colli, in mm^2^) and multifidus (rCSA-Multifidus, in mm^2^), as well as percentage of fat in muscle area (muscle fat infiltration, MFI) of longus colli (%MFI-CSA-Colli) and multifidus (%MFI-CSA-Multifidus) at C3/4, C4/5, C5/6, and C6/7 using the following formulas:
fat-free Longus colli CSA (rCSA-Longus colli) = (CSALongus colli) − (Longus colli fat)
fat-free Multifidus CSA (rCSA-Multifidus) = (CSAMultifidus) − (Multifidus fat)
%MFI-CSA-Multifidus = (CSAMultifidus fat/CSAMultifidus) × 100
%MFI-CSA-Colli = (CSAColli fat/CSA Colli) × 100


Multifidus and longus colli muscle volumes were calculated by a 3D multiplanar reconstruction of the muscles using Osirix (Figure 3) [45]. They were based on CSA measurements obtained on each slice every 3 mm from C4 to C7 for the multifidus and from C3 to C7 for the longus colli muscle. Fat-free muscle volume of longus colli (rVOL-Colli, in mm3) and multifidus (rVOL-Multifidus, in mm^3^) as well as percentage of fat in muscle volume of longus colli (%MFI-VOL-Colli) and multifidus (%MFI-VOL-Multifidus) were determined using the following formulas:
rVOL-Multifidus = (Multifidus Volume) − (Volume of Fat in Multifidus)
rVOL-Longus colli = (Longus colli Volume) − (Volume of Fat in Longus colli)
% Fat Muscle volume (%MFI-VOL-Muscle) = (Volume of Fat in muscle)/(Muscle Volume) × 100


Because muscle volume can vary according to height, fat-free muscle volume was also normalized by the participant’s height for the multifidus (Norm-rVOL-Multifidus) and the longus colli (Norm-rVOL-Colli) with the following formula:
Normalized fat-free muscle volume = (Fat-free muscle volume)/(participant’s height).

### 2.7. Statistical Analysis

Statistical analysis was conducted using SPSS (IBM SPSS 23.0; IBM Corp. Armonk, NY, USA). Normal distribution was assessed using the Shapiro–Wilk test. Bivariate correlation analyses were performed between NDI and neck muscle parameters using Pearson or Spearman correlation analysis. Correlation coefficients were considered low if between 0.30 and 0.50, moderate if between 0.50 and 0.70, high if between 0.70 and 0.90, and very high if between 0.90 and 1.00 [46]. The level of statistical significance was set at *p* < 0.05. A correlation matrix was generated to examine the data for multicollinearity. After this preliminary analysis, a multiple linear regression analysis was conducted to assess the relationship between NDI and neck muscle parameters. Selection of the best fit model was determined by significant main effects and interaction effects providing an overall significant model F-statistic (*p* < 0.05) and adjusted *r*^2^.

## 3. Results

From 38 volunteers who were considered, 25 were included in this study (20 females and 5 males). Those not included (*n* = 13) had shoulder pain without cervical involvement *n* = 4, disability-related to low pain back *n* = 2, concomitant carpal tunnel syndrome *n* = 1, clinical depression *n* = 1, or MRI data not usable *n* = 5.

Participants (*n* = 25) had a mean age of 47.3 ± 9.6 years, height of 1.66 ± 0.1 m, and body mass index of 24.6 ± 5.2 kg/m^2^.

Participant clinical characteristics and measures of neck muscle function are presented in Table 1 and Table 2. MRI measures of multifidus cervical muscle cross-sectional area were 215.4 ± 50.3 mm^2^, 251.5 ± 51.3 mm^2^, 266.3 ± 54.9 mm^2^, and 348.1 ± 97.9 mm^2^ at C3/4, C4/5, C5/6, and C6/7 levels, respectively. MRI measures of cervical muscle cross-sectional area and volume are presented in Table 3 and Table 4.

The bivariate correlation analysis revealed that there were moderate correlations between NDI and %Fat Multifidus-Vol (*r* = 0.572; *p* = 0.003) and %Fat Multifidus-CSA at C6/7 (*r* = 0.552; *p* = 0.004), and low correlation between NDI and %Fat Multifidus-CSA at C4/5 (*r* = 0.497; *p* = 0.011) and C5/6 (*r* = 0.498; *p* = 0.011) (Table 5). There were moderate and negative correlation between NDI and rCSA-Multifidus at C6/7 (*r* = −0.548; *p* = 0.005).

There were no correlations between the NDI and any measures of longus colli.

Multiple linear regression did not include CSA variables so as to avoid collinearity with their respective volume variables. As expected, CSA measurements (C3/4, C4/5, C5/6, C6/7) were significantly correlated to their respective muscle volume variables (%MFI-Multifidus at C3/4, C4/5, C5/6, C6/7, and %MFI-Vol-Multidus). Multiple linear regression analysis with backward elimination method was performed with NDI as the independent variable, and CCFT, neck flexor and extensor endurance, %MFI-Vol-Multifidus, %MFI-Vol-Colli, Norm-r-Vol-Multifidus, and Norm-r-Vol-Colli as dependent variables. One model was significant and obtained the best adjusted *r*^2^ (adjusted *r*^2^ = 0.317). In this model, NDI had a significant positive correlation with %MFI-Vol-Multifidus (B = 0.496) and a negative correlation with Norm-r-Vol-Multifidus (B = −0.230) (*p* = 0.006). Together, the variance of both parameters explained 32% of the variance of NDI.

## 4. Discussion

This study examined the relationship between neck disability severity as evaluated by the Neck Disability Index (NDI) and cervical muscle morphology and composition (fat and fat-free muscle volumes) and function (motor control and muscle endurance) in patients with CINP. We hypothesized that the majority of NDI variance would be explained by these measures.

This study determined that the NDI score was moderately and significantly correlated with the percentage of multifidus volume fat infiltration (%Fat Multifidus-Vol). There was an overall low correlation between the NDI score and the volume of the multifidus after fat had been removed and the measure adjusted for height (Norm-rVOL-Multifidus). Together, these parameters explained 32% of the variance of NDI.

Our measures of multifidus CSA were similar to those reported in persons with whiplash-associated disorders (270–290 mm^2^ at C5) [31] and in patients prior to cervical spine decompression surgery (294.2 mm^2^ at C5/6) [47]. Likewise, our measures of multifidus fat infiltration were similar to those documented in whiplash-associated disorders (17–29.8%) [28,31] and in patients prior to cervical spine decompression surgery (31.7%) [25]. Fat infiltration in the cervical multifidus muscle has been well documented, notably in persons with whiplash-associated disorders who had poor recovery after injury and higher disability [26,28,31,48], as well as in patients with degenerative cervical myelopathy with poor functional recovery after surgery [30]. While the findings of our study in patients with CINP concur with these observations, they contrast with the results of Elliott et al. [17], who found that cervical multifidus fat infiltration at C5/6 in their cohort of patients with CINP was not significantly different from asymptomatic participants. This disparity might reflect the heterogeneity of neck pain disorders within this very broad classification. The cohort of patients with CINP of Elliott et al. [17] was recruited from advertising in the community (NDI = 21.9% ± 7.5), whereas our cohort included patients who had been referred to a neurosurgery department (NDI = 38.6% ± 12.3). One could speculate that our participants had more advanced pathologies akin to patients with poor recovery from whiplash injury (NDI = 45.5% ± 15.8) [17].

For the longus colli muscle, our study determined that the NDI was unrelated to any fat or volume measures pertaining to the longus colli muscle. Our values obtained for neck flexor muscle CSA were similar to those reported for longus capitis for CINP and WAD [17], but fat infiltration of longus capitis was lower [17]. It is possible that the deep flexor muscles are less readily subject to this change compared to the cervical multifidus.

The results of our study also indicated no relationship between NDI scores and neck flexor or extensor muscle endurance, or the outcome of the low-load motor control test of craniocervical flexion. These results are similar with the discrepancy found by Martins et al. [11]. Endurance or low-load motor control tests were not associated with either CSA or MFI of muscles.

Multiple linear regression analysis indicated that NDI was positively correlated with the volume percentage of fat infiltration of the multifidus (B = 0.496), negatively correlated with fat-free muscle volume of the multifidus normalized by subject height (B = −0.230), and accounted for 32% of the variance. A negative association in patients with CINP between the NDI and the cross-sectional area of semispinalis cervicis (a deep muscle closely positioned to the multifidus) was also found in an ultrasound imaging study [14]. Together, these findings suggest that while both morphometry and composition of multifidus may influence NDI, there are undoubtedly many other factors across physical, physiological, and psychological domains that may also contribute to the NDI score. Nevertheless, the development and magnitude of fat infiltration in multifidus cervical muscle has been shown to be an important predictor of poor outcomes following a whiplash injury [28]. It is unknown as to whether these quantitative measures would be helpful in predicting the progression of disease in CINP, but it is a reasonable area for future research. Winslow et al. [49] demonstrated the predictive value of the measure of fat infiltration of the lumbar multifidus for return to sport in young athletes with extension-based lower back pain. treatment.

We measured the fat infiltration in the multifidus and longus colli muscles at four cervical segments, from C3/4 to the C6/7, in our participants with CINP. Fat infiltration was greatest at the C6/7 (41.1%) and C3/4 (36.9%) levels and lesser at the C5/6 (25.5%) and C4/5 (20.6%) levels. This distribution is not dissimilar to that documented by Elliott et al. [50] in a cohort with whiplash-associated disorders. It is possible that this distribution may relate to the frequency or magnitude of pathologies at the respective segments. Interestingly, recent studies have shown greater fat infiltration in the cervical multifidus in patients with whiplash-associated disorders compared with healthy persons, but no difference in fat infiltration in the lower limb muscles [51,52]. This suggests that fat infiltration in multifidus is a local effect rather than a systemic effect of age or activity. Experimental studies of the lumbar region in animal and human models have found that local inflammatory processes occur specifically in deep posterior spine muscles, which increases the muscle fiber transition and intramuscular fat in the multifidus muscle [53,54]. Similar mechanisms could explain the fat infiltration in the cervical multifidus in patients with CIPN, but this is yet to be verified in a controlled environment. In a longitudinal study of patients with acute whiplash injury (<3 weeks) [55], it was shown that there was a moderate negative correlation between serum inflammatory biomarker levels (notably TNF-α) and amount of fatty muscle infiltrate in the cervical extensor muscles at 3 months. Hence, both local and systemic mechanisms need to be researched to fully understand the reasons for and differing magnitudes of morphological change in the deep cervical muscles. Theses findings may have direct clinical implications for the management of chronic non-specific neck pain by prescribing specific exercises which could contribute to changing muscle fat infiltration.

## 5. Strengths and Limitations

A strength of our study was that we measured the volume of fat infiltration in neck muscles. This improves 2D methods that may suffer from errors of measurement associated with partial volume effect [56]. A recent systematic review advocated for 3D muscle morphometry (used in this study) to be the standard method to determine if fat infiltration is a relevant marker for chronic neck pain [57]. Our measures are an average between sides, a method supported by Yun et al. [58] who found no significant difference in neck muscle CSA or relative muscle CSA (relative fat infiltration) between the affected and unaffected side in patients with chronic cervical radiculopathy.

A clear limitation of our study was the neglect to measure all flexor and extensor muscles (both deep and superficial) as well as serum inflammatory biomarker levels (notably TNF-α) [55] or microRNA let-7i-5p [59] in exploring the relationship between muscle function measures and morphological change. Future studies might also examine the relationship between muscle morphological change and other factors such as psychological states.

The angles of cervical lordosis were also not considered in this study and may be a consideration in future research as relationships are very uncertain. A correlation has been found between semispinalis cervicis CSA and loss of cervical lordosis [60,61], but 2D-measurement used in these studies may lead to errors of measurement associated with partial 3-D volume effect. The loss of cervical kyphosis leads to increasing the length and decreasing the CSA of the neck extensors. Studies found that cervical lordosis was correlated with fat infiltration of deep neck extensors (multifidus, spinalis cervicis, and capitis) at the C4/5 and C7/T1 segments [62] but not at C5/6 [63]. Xing-Jin Wang et al. [64] found no differences in cervical lordosis angle in patients with more extensor muscle fat infiltration. Future research is needed to investigate the effect of disk degeneration on spinal alignment and muscle composition.

## 6. Conclusions

This study examined the relationship between severity of neck disability reported by participants with chronic non-specific neck pain and cervical muscle structure (atrophy, fat infiltration) and function (motor control and muscle endurance). Neck disability was moderately correlated with the percentage of fat volume in the multifidus muscle and fat-free volume of the multifidus, and it explained 32% of the variance. Neck disability was unrelated to any fat or volume measures pertaining to the longus colli muscle. There was no relationship between NDI scores and muscle function test outcomes.

## Figures and Tables

**Figure 1 jcm-11-05522-f001:**
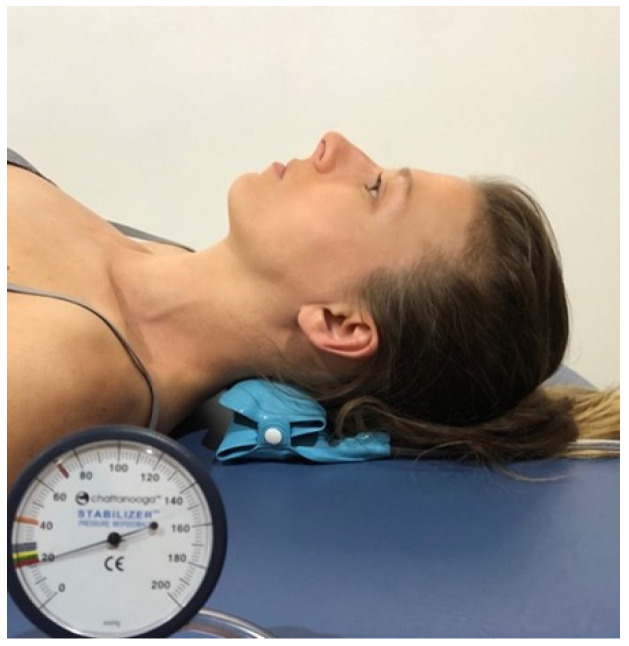
Craniocervical flexion test.

**Figure 2 jcm-11-05522-f002:**
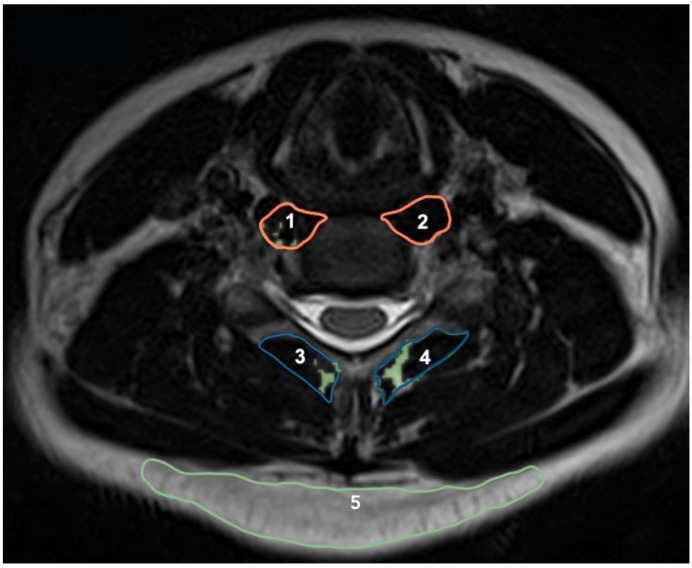
Measurement of cross-sectional area of the longus colli (1 and 2), multifidus (3 and 4), fat of posterior subcutaneous (5), and muscle fat infiltration (in green) at the C5-C6 level.

**Figure 3 jcm-11-05522-f003:**
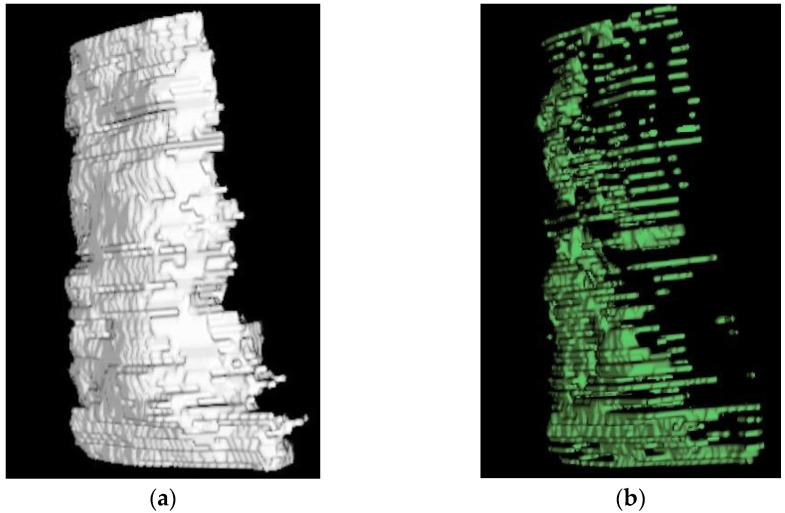
A three-dimensional multiplanar reconstruction of the multifidus muscle (**a**) and fat infiltration of the multifidus muscle (**b**).

**Table 1 jcm-11-05522-t001:** Clinical characteristics of the participants (*n* = 25).

Clinical Characteristics	Mean (SD)	Min–Max
Duration (months)	77.1 (76.3)	4–240
Numeric Pain Rating Scale	5.1 (1.6)	3–8
Neck Disability Index (NDI)	38.6 (12.3)	20–66
Pain Catastrophizing Scale	21.9 (10.8)	3–37
PCS-12	35.3 (6.1)	27.4–46.3
MCS-12	34.2 (7.4)	23.7–56.4
Cervical flexion ROM (°)	50.6 (16.4)	20–80
Cervical extension ROM (°)	46.5 (17.6)	5–100
Right cervical lateral flexion ROM (°)	29.7 (10.2)	12–52
Left cervical lateral flexion ROM (°)	29.4 (9.4)	13–56
Right cervical rotation ROM (°)	59.7 (9.2)	39–86
Left cervical rotation ROM (°)	61.0 (10.0)	43–90

SD: standard deviation; ROM: range of motion; PCS-12: The Physical Component of the Short-Form 12-Item Health Survey; MCS-12: The Mental Component of the Short-Form 12-Item Health Survey.

**Table 2 jcm-11-05522-t002:** Functional measures of the neck muscles.

Neck Muscle Parameters	Mean (SD)	Min–Max
CCFT (mmHg)	23.6 (1.7)	20.0–28.0
Endurance neck flexors (s)	20.0 (10.1)	6.3–57.7
Endurance neck extensors (s)	36.8 (22.4)	8.4–114.5

SD: standard deviation; CCFT: Craniocervical Flexion Test; s: seconds; mmHg: millimeter of mercury.

**Table 3 jcm-11-05522-t003:** MRI measures of the neck muscles’ cross-sectional areas.

		**Mean (SD)**	**Min–Max**
**CSA C3/4**	rCSA-Multifidus (mm^2^)	132.5 (37.1)	51.9–197.1
	%MFI-CSA-Multifidus	36.9 (17.4)	2.8–69.7
	rCSA-Colli (mm^2^)	124.1 (23.2)	96.7–187.3
	%MFI-CSA-Colli	0.1 (0.1)	0.0–0.2
**CSA C4/5**	rCSA-Multifidus (mm^2^)	198.0 (39.5)	135.9–288.4
	%MFI-CSA-Multifidus	20.6 (9.4)	7.0–41.8
	rCSA-Colli (mm^2^)	145.7 (38.3)	82.5–214.1
	%MFI-CSA-Colli	6.0 (6.7)	0–21.9
**CSA C5/6**	rCSA-Multifidus (mm^2^)	196.3 (43.3)	124.4–300.8
	%MFI-CSA-Multifidus	25.5 (11.6)	9.1–47.2
	rCSA-Colli (mm^2^)	134.2 (36.3)	66.4–205.4
	%MFI-CSA-Colli	4.9 (5.2)	0–20.4
**CSA C6/7**	rCSA-Multifidus (mm^2^)	201.0 (78.5)	51.3–406.6
	%MFI-CSA-Multifidus	41.1 (18.3)	15.2–85.2
	rCSA-Colli (mm^2^)	163.2 (31.8)	103.4–214.2
	%MFI-CSA-Colli	4.4 (7.4)	0–30.5

SD: standard deviation. CSA: cross-sectional area. rCSA-Multifidus (mm^2^): fat-free muscle cross-sectional area of multifidus. %MFI-CSA-Multifidus: percentage of fat infiltration in multifidus cross-sectional area. rCSA-Colli (mm^2^): fat-free muscle cross-sectional area of longus colli. %MFI-CSA-Colli: percentage of fat infiltration in longus colli muscle cross-sectional area.

**Table 4 jcm-11-05522-t004:** MRI volume measures of the neck muscles.

Volume Measurements	Mean (SD)	Min–Max
VOL-Multifidus (mm^3^)	12,953.0 (3243.9)	8795.5–23,892.7
VOL-Colli (mm^3^)	10,299.0 (2390.9)	5950.9–15,418.5
rVOL-Multifidus (mm^3^)	9146.8 (2322.6)	6760.8–17,506.0
rVOL-Colli (mm^3^)	9100.1 (2136.5)	5434.5–13,988.4
Norm-rVOL-Multifidus	54.7 (11.1)	41.0–90.6
Norm-rVOL-Colli	54.5 (10.6)	33.7–78.6
%MFI-VOL-Multifidus (%)	28.6 (9.3)	11.58–52.0
%MFI-VOL-Colli (%)	11.5 (5.1)	4.3–22.4

SD: standard deviation. VOL-Multifidus: volume of multifidus. VOL-Colli: volume of longus colli. rVOL-Multifidus: fat-free volume of multifidus. rVOL-Colli: fat-free volume of longus colli. Norm-rVOL-Multifidus: normalized fat-free volume of multifidus. Norm-rVOL-Colli: normalized fat-free volume of longus colli. %MFI-VOL-Multifidus: percentage of fat infiltration in multifidus volume. %MFI-VOL-Colli: percentage of fat infiltration in longus colli volume.

**Table 5 jcm-11-05522-t005:** Correlation between muscle parameters and neck disability index.

Neck Muscle Parameters	Coefficient of Correlation with NDI	
CCFT (mmHg) °	0.025	*p* = 0.905
Endurance neck flexors (s) °	−0.093	*p* = 0.657
Endurance neck extensors (s) °	0.032	*p* = 0.878
**Volume**		
Norm-rVOL-Multifidus °	−0.380	*p* = 0.061
Norm-rVOL-Colli	0.124	*p* = 0.555
%MFI-VOL-Multifidus	0.572	*p* = 0.003 *
%MFI-VOL-Colli	0.195	*p* = 0.349
**CSA C3/4**		
rCSA-Multifidus (mm^2^)	−0.047	*p* = 0.823
%MFI-CSA-Multifidus	0.313	*p* = 0.128
rCSA-Colli (mm^2^) °	0.130	*p* = 0.534
%MFI-CSA-Colli °	0.181	*p* = 0.385
**CSA C4/5**		
rCSA-Multifidus (mm^2^)	−0.117	*p* = 0.578
%MFI-CSA-Multifidus	0.497	*p* = 0.011 *
rCSA-Colli (mm^2^)	−0.061	*p* = 0.771
%MFI-CSA-Colli °	−0.038	*p* = 0.857
**CSA C5/6**		
rCSA-Multifidus (mm^2^)	−0.365	*p* = 0.073
%MFI-CSA-Multifidus °	0.498	*p* = 0.011 *
rCSA-Colli (mm^2^)	0.057	*p* = 0.787
%MFI-CSA-Colli °	−0.228	*p* = 0.273
**CSA C6/7**		
rCSA-Multifidus (mm^2^)	−0.548	*p* = 0.005 *
%MFI-CSA-Multifidus °	0.552	*p* = 0.004 *
rCSA-Colli (mm^2^)	0.347	*p* = 0.089
%MFI-CSA-Colli °	−0.038	*p* = 0.123

rCSA-Multifidus (mm^2^): fat-free muscle cross-sectional area of multifidus. %MFI-CSA-Multifidus: percentage of fat infiltration in multifidus cross-sectional area. rCSA-Colli (mm^2^): fat-free muscle cross-sectional area of longus colli. %MFI-CSA-Colli: percentage of fat infiltration in longus colli muscle cross-sectional area. Norm-rVOL-Multifidus: normalized fat-free volume of multifidus. Norm-rVOL-Colli: normalized fat-free volume of longus colli. %MFI-VOL-Multifidus: percentage of fat infiltration in multifidus volume. %MFI-VOL-Colli: percentage of fat infiltration in longus colli volume. ° Spearman coefficient. * *p* < 0.05.

## Data Availability

The data are not publicly available due to privacy or ethical restrictions. Data belongs to the local hospital research. The findings of this study are available from the corresponding author upon reasonable request.

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
