# Peer review of "Fat Infiltration of Multifidus Muscle Is Correlated with Neck Disability in Patients with Non-Specific Chronic Neck Pain"

_jcm, 2022, doi:10.3390/jcm11195522_

Round 1

Reviewer 1 Report

The paper by Grondin et al. entitled “Fat infiltration of multifidus muscle is correlated with neck disability in patients with idiopathic chronic neck pain." is an interesting study on pathophysiological mechanism of chronic idiopathic chronic pain. The authors examined neck disability and cross-sectional area, volume and composition of cervical muscle acquired from cervical MRI. They investigated the relationship between severity of neck disability of patient with chronic idiopathic neck pain, muscle structure and function.

They found that Neck Disability Index (NDI) was moderately positively correlated with mutifidus muscle fat infiltration, and negatively correlated to normalized fat-free muscle volume of multifidus. They reported there was no relationship between NDI score and neck flexor and extensor muscle endurance and motor control. They concluded that neck muscle fat infiltration of multifidus influenced NDI without any relationship between muscle endurance and NDI.

In general, the data is clearly presented and will be of interest to pain physician, Orthopaedic Surgeon and Neurosurgeon.

I have the following concerns.

Major comment

(1)    In the “Materials and Method” section, the authors mentioned they investigated thirty-three patients (Page 3 line.2). However, they reported twenty-five patients were examined in “Results” section (Page5, line2 in Results). How the authors excluded the eight patients? They should show the exclusion criteria of this matter.

(2)    The authors investigated patients with chronic idiopathic neck pain, who sustained neck pain more than 3 months. In such patients, biopsychosocial factors may influence their symptoms. According to previous report, cut-off value of pain catastrophizing scale is over 30 points, and that of mental component of SF-12 is between 49 and 50. In this study, the author included the patients who showed the score points over cut-off values of these scores. They had better exclude these patients.

(3)    Page6, line 5-6

The authors mentioned that thirteen volunteers were excluded. I wonder whether they excluded 13 volunteers out of 33 participants, or out of 25 participants. Considering this definition, should readers understand Table 3,4 show data acquired from 20 patients, or 12 patients?

(4)    The authors concluded that there was no relationship between NDI and muscle endurance or motor control. NDI is comprehensive evaluation tool for neck disability. Pain intensity can influence activity of daily living of chronic pain patients. They should pay attention to the relationship between pain intensity and muscle endurance, or % fat infiltration of multifidus or longus colli muscle. Subgroup analysis of NDI, especially for section one (pain intensity) would be helpful.

(5)    The author focused on localized inflammation of posterior cervical muscle. Spinal alignment, such as cervical kyphosis or hyper-lordosis could affect posterior cervical muscle. They had better argue factors of spinal alignment in Discussion section.

Minor comments

(1)    Page 6, line 2-3

The value of multifidus CSA at C6/7 was missing. The author should additionally describe the value of this.

(2)    The authors defined chronic “idiopathic” neck pain as the neck pain without any specific pathological causes. The term of “idiopathic” might be awkward. It should be revised. Non-specific neck pain would be better.

(3)    Cranio-cervical flexion test might not be popular for readers in clinical setting. The authors cited previous report on this test. However, photograph of this test would be helpful for readers.

Author Response

The paper by Grondin et al. entitled “Fat infiltration of multifidus muscle is correlated with neck disability in patients with idiopathic chronic neck pain." is an interesting study on pathophysiological mechanism of chronic idiopathic chronic pain. The authors examined neck disability and cross-sectional area, volume and composition of cervical muscle acquired from cervical MRI. They investigated the relationship between severity of neck disability of patient with chronic idiopathic neck pain, muscle structure and function. 

They found that Neck Disability Index (NDI) was moderately positively correlated with mutifidus muscle fat infiltration, and negatively correlated to normalized fat-free muscle volume of multifidus. They reported there was no relationship between NDI score and neck flexor and extensor muscle endurance and motor control. They concluded that neck muscle fat infiltration of multifidus influenced NDI without any relationship between muscle endurance and NDI.

In general, the data is clearly presented and will be of interest to pain physician, Orthopaedic Surgeon and Neurosurgeon.

I have the following concerns.

Major comment

(1)    In the “Materials and Method” section, the authors mentioned they investigated thirty-three patients (Page 3 line.2). However, they reported twenty-five patients were examined in “Results” section (Page5, line2 in Results). How the authors excluded the eight patients? They should show the exclusion criteria of this matter.
We apologise for these mistakes.We deleted this part in the “Materials and Method” section and added more detailed information in the results section as suggested below. We thank the reviewer for this remark.

(3)    Page6, line 5-6

The authors mentioned that thirteen volunteers were excluded. I wonder whether they excluded 13 volunteers out of 33 participants, or out of 25 participants. Considering this definition, should readers understand Table 3,4 show data acquired from 20 patients, or 12 patients?
Thank you for highlighting the misunderstanding. This was clarified in the article.
“From thirty-eight volunteers who were considered, twenty-five were included in this study (20 females and 5 males). Patients who consulted but were not included in the study (n=13) had shoulder pain without cervical involvement n=4, disability-related to low pain back n=2, concomitant carpal tunnel syndrome n=1, clinical depression n=1, or if MRI were not usable n=5. Participants (n=25) had a mean age of 47.3±9.6 years, height of 1.66±0.1 meters, body mass index of 24.6±5.2 kg/m²”
p.6  line 24-27

(4)    The authors concluded that there was no relationship between NDI and muscle endurance or motor control. NDI is comprehensive evaluation tool for neck disability. Pain intensity can influence activity of daily living of chronic pain patients. They should pay attention to the relationship between pain intensity and muscle endurance, or % fat infiltration of multifidus or longus colli muscle. Subgroup analysis of NDI, especially for section one (pain intensity) would be helpful.
Thank you for this comment and the discussion of the mechanisms involved in the changes in cervical muscle composition, morphometry and function. Although the study of these mechanisms is of great interest, we think that these analyses should be presented in another article so as not to complicate this one.

(5)    The author focused on localized inflammation of posterior cervical muscle. Spinal alignment, such as cervical kyphosis or hyper-lordosis could affect posterior cervical muscle. They had better argue factors of spinal alignment in Discussion section.
Thank you for this comment, we have added the following to the limitations section :
The angles of cervical lordosis were also not considered in this study and maybe a consideration in future research as relationships are very uncertain. A correlation has been found between semispinalis cervicis CSA and loss of cervical lordosis [60,61] but 2D-measurement used in these studies may lead to errors of measurement associated with partial 3-D volume effect. The loss of cervical kyphosis leads to increasing the length and decreasing the CSA of the neck extensors. Studies found that cervical lordosis was correlated with fat infiltration of deep neck extensors (multifidus, spinalis cervicis and capitis) at the C4/5 and C7/T1 segments [62] but not at C5/6 [63]. Xing
jin Wang [64] found no differences in cervical lordosis angle in patients with more extensor muscle fat infiltration. Future research is needed to investigated the effect of disk degeneration on spinal aligment and muscle composition.
p.11 line 22

Minor comments

(1)    Page 6, line 2-3
The value of multifidus CSA at C6/7 was missing. The author should additionally describe the value of this.
We added the multifidus CSA value missing “251.5±51.3mm2” p.6 line 32. Thank you.

(2)   The authors defined chronic “idiopathic” neck pain as the neck pain without any specific pathological causes. The term of “idiopathic” might be awkward. It should be revised. Non-specific neck pain would be better.
We agree with the reviewer. Idiopathic has been changed to non-specific throughout the text.

(3)    Cranio-cervical flexion test might not be popular for readers in clinical setting. The authors cited previous report on this test. However, photograph of this test would be helpful for readers.
Following the reviewer remark, a photograph has been added. Please see Figure 1. p.4

We thank the reviewer for his valuable comments which have improved the quality and comprehensibility of our article.

Reviewer 2 Report

1) Why were patients send to the Neurosurgery Department? I expect them to have severe problems due to the need to go there?

2 ) I am missing explanations of the questionnaires you used. If the NDI is one of your major outcomes (disability) explain it better.

3) Please check your number of participants. You have different numbers throughout the paper.

4) What are SF-12 and Pain Catastrophizing Scale relevant for in your research  question? Why did you collect this data?

5) You would like to predict the NDI right? Then it should be your dependent variable? What happened to SF-12 and Pain Catastrophizing Scale?

Author Response

1) Why were patients send to the Neurosurgery Department? I expect them to have severe problems due to the need to go there?

The service has extended the reasons for consultations for research purposes by seeing patients who do not require surgery. In addition, the head of this department is Dr Freppel, who is also an investigator in the study. For these reasons, the patients consulted the doctor in this department.

2 ) I am missing explanations of the questionnaires you used. If the NDI is one of your major outcomes (disability) explain it better.
The reviewer is right, it is indeed important to describe this questionnaire in more detail. We added this p.3 line8
The NDI is the most widely used assessment tool measuring disability in patients with acute and chronic neck pain or neck injury. It has been shown as an efficient and trustworthy tool to measure and monitor neck-related disability [33]. The NDI contains 10 items; pain, personal care, lifting, reading, headaches, concentration, work, driving, sleeping and recreation. Each item is scored on a 0 to 5 rating scale. The NDI is scored with points summed /50 and expressed as a percent: 0-4points (0-8%) no disability, 5-14 points (10–28%) mild disability, 15-24points (30-48%) moderate disability, 25-34points (50-64%) severe disability, 35-50points (70-100%) complete disability”.
p.3 line13

3) Please check your number of participants. You have different numbers throughout the paper.
We apologize, we have corrected this in the paper p6 line24
« From thirty-eight volunteers who were considered, twenty-five were included in this study (20 females ; 5 males). Those not included (n=13) had shoulder pain without cervical involvement n=4, disability-related to low pain back n=2, concomitant carpal tunnel syndrome n=1, clinical depression n=1, or if MRI were not usable n=5.
Participants (n=25) had a mean age of 47.3±9.6 years, height of 1.66±0.1 meters, body mass index of 24.6±5.2 kg/m² »

4) What are SF-12 and Pain Catastrophizing Scale relevant for in your research question? Why did you collect this data? The SF-12 and Pain Catastrophizing Scale were not relevant for the research question.
Indeed, the data of Physical, Mental Component of the SF-12 and Pain Catastrophizing Scale are not relevant to directly answer the research question, but we have included these data to provide a baseline characterization of our sample. We believe that this information is useful to properly describe the patients included in this study, but if requested by the reviewer, we will remove the data.

5) You would like to predict the NDI right? Then it should be your dependent variable? What happened to SF-12 and Pain Catastrophizing Scale?

As mentioned above, we have included the SF12 and PCs to provide a baseline characterization of our sample.
As proposed above, if requested by the reviewer or the editor, we can remove the data if they distract from the main aim of this study.

We thank the reviewer for his valuable comments which have improved the quality and comprehensibility of our article.

Round 2

Reviewer 1 Report

Based on the reviewer’s comments, the authors properly revised. The authors added figure which shows muscle examination. It would be helpful for readers.

Considering review of the revision of this article, I would like to recommend to editor as acceptable paper.

Reviewer 2 Report

None